# AIF-GUIDED DIFFUSION PLANNING FOR NONSTATIONARY CONTROL

## ABSTRACT

We present AIF-guided diffusion planning for rapid, reward-free adaptation to abrupt, within-episode dynamics shifts in continuous control. Our controller couples *Active Inference* (AIF)—which maintains a compact belief over latent dynamics and pursues actions that minimize *Expected Free Energy* (EFE), balancing goal pursuit and information gain—with a conditional trajectory diffusion policy conditioned on state, goal, and a compact embedding of the regime belief. At deployment, prediction surprise under a hazard prior triggers Bayesian changepoint resets and fast belief correction; the updated belief both guides denoising and supplies an EFE-based scorer over candidate plans, executing the first action of the minimum-EFE proposal in closed loop. Trained offline on regime-stratified datasets (no cross-regime transitions), the method adapts without test-time rewards or ensembles. On nonstationary MuJoCo HalfCheetah with mid-episode gravity switches, it improves mid-window reward (steps 100–200) and preserves final return relative to strong offline baselines. An unsupervised variant that discovers regimes from trajectories remains robust, though slower to recover; ablations indicate the belief pathway and EFE guidance drive the gains.

## 1 INTRODUCTION

Real systems rarely run on a single, tidy set of equations. A manipulator grasps a heavier part, a wheel heats and grip rises, a crosswind nudges a quadrotor off course. Friction, mass, payload, weather, and wear shift within a single episode and the mapping from action to outcome drifts. The stationarity assumed by many planners and offline policies then breaks down. In safety-critical applications such as autonomous driving and mobile manipulation, the controller must update its understanding on the fly and do so without sacrificing control quality (Peng et al., 2018; Kumar et al., 2021). A convenient formal device makes this concrete: introduce a latent regime variable $z_t$ that modulates the transition dynamics so nonstationarity becomes inference over $z_t$ (Doshi-Velez & Konidaris, 2013).

A large body of work seeks to adapt under these conditions, but with different levers. Gradient-based meta-learning and recurrent meta-RL compress experience into parameters or state so a few fresh interactions can re-tune behavior quickly (Finn et al., 2017; Duan et al., 2017). Probabilistic context methods maintain a posterior over a latent task or regime and condition policies on that belief (Rakelly et al., 2019; Zintgraf et al., 2020). In practice, many of these pipelines depend on reward during adaptation and can become brittle when the switch occurs mid-trajectory. Others trade robustness for heavy computation by using ensembles or costly test-time optimization, increasing latency when decisions are time-critical.

Active Inference (AIF) offers a different organizing principle. Cast control as inference, maintain beliefs over latent and world states, and choose actions by minimizing Expected Free Energy (EFE)—a quantity that balances goal-directed behavior (expected utility) with information-seeking actions (epistemic value) in a single objective (Friston, 2010; Parr & Friston, 2019; Millidge et al., 2021). The appeal is conceptual simplicity and a built-in drive to resolve uncertainty about the current regime. The challenge is engineering. Accurate generative models are hard to learn, and naively evaluating EFE across candidate futures is expensive, which limits scalability in continuous control.

Denoising diffusion models complement these strengths. They produce diverse, coherent trajectory samples and admit efficient test-time guidance that can steer sampling toward task-relevant futures

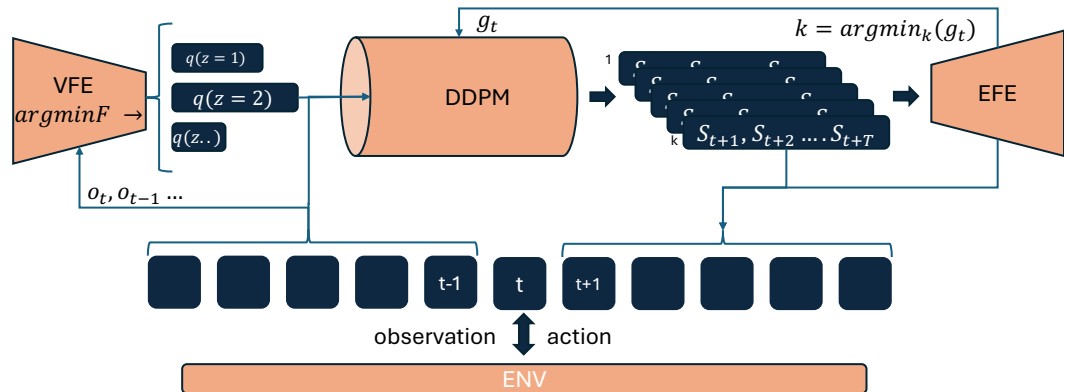

Figure 1: **Belief-conditioned diffusion planning with EFE guidance.** VFE updates $q(z_t)$ and state from observations under a hazard prior; a DDPM samples $K$ horizon-$H$ plans conditioned on $(s_t, g, \tilde{z}_t)$; EFE provides gradients to guide denoising and scores to select $k^\star$, whose first action is executed. The environment returns $o_{t+1}$, closing the loop.

(Sohl-Dickstein et al., 2015; Ho et al., 2020; Dhariwal & Nichol, 2021; Ho & Salimans, 2022). In offline control, diffusion planners denoise entire trajectories conditioned on state and goals, delivering stable performance without online gradient steps (Janner et al., 2022; Ajay et al., 2023). Yet most existing planners assume stationary dynamics at test time (Brehmer et al., 2023; Chen et al., 2024). As a result they generate plausible plans for the wrong world when a hidden shift occurs, precisely when fast, belief-aware redirection is needed.

This paper targets rapid, reward-free adaptation to abrupt within-episode dynamics shifts by combining these ideas. We couple a fast variational belief updater with a conditional diffusion trajectory planner trained offline on regime-specific datasets (no cross-regime transitions required). At deployment, the controller executes the loop in Fig. 1: (i) given observations $o_{\leq t}$, a variational free-energy (VFE) objective updates the posterior $q(z_t \mid o_{\leq t})$ and a filtered state; prediction surprise, compared to a hazard prior, detects changepoints and resets belief when needed; (ii) a diffusion model samples $K$ horizon-$H$ trajectories conditioned on the current state $s_t$, goal $g$, and a compact embedding $\tilde{z}_t$ of the regime belief; (iii) a lightweight EFE functional supplies guidance during denoising and then scores the $K$ candidates, with the first action of the minimum-EFE plan executed; (iv) the environment returns $o_{t+1}$ and the cycle repeats. This design keeps adaptation reward-free, avoids ensembles, and yields fast recovery after a switch, while preserving return relative to strong offline RL and diffusion baselines on nonstationary MuJoCo tasks.

Empirically, we evaluate the controller (AIDIF) in a nonstationary HalfCheetah protocol with 300-step rollouts: default dynamics for the first 100 steps, a gravity increase to 5×at t=100, and reversion at t=200, with state carried across changes. We report **Return@100**, **Reward 100–200**, and **Return@300**; unless noted, returns are D4RL-normalized and the 100–200 reward window uses the same normalized units. Offline training mixes D4RL `halfcheetah-medium-v2` with off-dynamics SAC logs spanning gravity, friction, and morphology variants. AIDIF with regime labels attains Reward 100–200 236.85 vs 8.58 for Decision Diffuser and -52.85 for IQL, and Return@300 1100.56 vs 825.11 and 89.88, while matching pre-shift competence at Return@100. An unsupervised variant that infers regimes without labels remains robust but trails the labeled model. Ablations over hazard rate, guidance activation, candidate count, and regime tagging indicate that the belief update and EFE-guided denoising are the dominant levers. These results support that explicit dynamics inference plus belief-conditioned diffusion yields faster post-switch recovery than stationary diffusion or value-based baselines under equal training budgets (Fig.3, Table1).

**Contributions.** We make three primary contributions: (1) We introduce a belief-conditioned diffusion planning framework that integrates active inference principles with modern generative models for rapid adaptation to within-episode dynamics changes without requiring reward signals during test time; (2) We demonstrate that coupling surprise-based changepoint detection with EFE-guided tra-

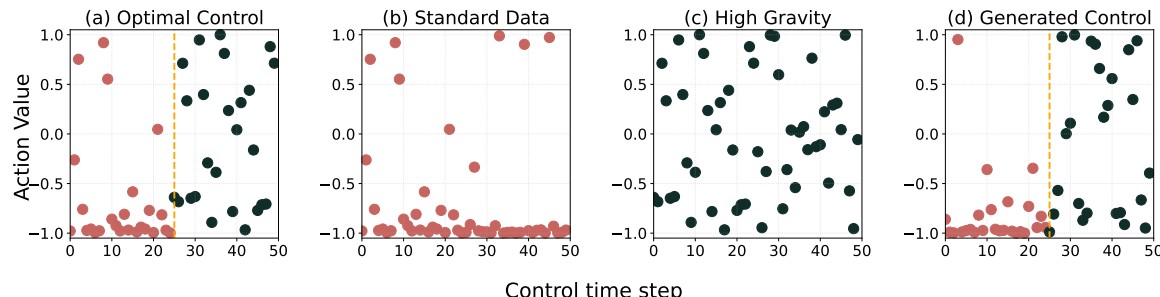

Figure 2: **Intuition.** One episode and one hidden switch for exposition. (a) Optimal actions before and after the change; the dashed line marks the unknown switch time. (b)–(c) Stationary action distributions in the two regimes. (d) Generated control that shifts as beliefs update.

jectory generation enables faster recovery from abrupt regime shifts compared to existing offline RL and diffusion baselines; (3) We provide both supervised and unsupervised variants of our approach, showing that explicit dynamics inference improves robustness even without ground-truth regime labels, with comprehensive ablations identifying the belief update mechanism and EFE guidance as the key components driving performance gains.

## 2 BACKGROUND

Real systems switch regime within an episode. Friction, payload, or gravity can change so the same command yields different motion; a stationary policy then degrades. Figure 2 anchors the problem. Panel (a) shows one episode with a latent change near $t \approx 25$. Before the change, small actions are optimal (red); after, larger actions are needed (green). Panels (b)–(c) show the stationary action distributions under two regimes. Panel (d) is the goal: as beliefs update, the sampler should move its outputs from (b) to (c) quickly while staying task-directed. We model episodes with a discrete latent regime that modulates dynamics (Doshi-Velez & Konidaris, 2013) and use fast test-time adaptation (Peng et al., 2018; Kumar et al., 2021). Objective: rapid, reward-free recovery after a hidden switch.

**Notation.** States $s_t \in \mathcal{S}$, actions $a_t \in \mathcal{A}$, observations $o_t \in \mathcal{O}$. Discrete regime $z_t \in \{1, \ldots, |\mathcal{Z}|\}$. Horizon $H$. History $h_t = (o_{\leq t}, a_{<t})$. $p(\cdot)$ is the generative model; $q(\cdot)$ is the online belief. Changepoint hazard $h \in (0, 1)$ induces a geometric run-length prior. Preferences use a goal likelihood $p(o_t^\star \mid s_t)$; no reward is used at test time. For diffusion planning the horizon-$H$ plan is $x_0 = [a_t, \ldots, a_{t+H-1}] \in \mathbb{R}^{dH}$. Diffusion time is $n$.

### 2.1 PROBLEM SETUP

We use a POMDP whose dynamics depend on $z_t$; the joint factorizes as

$$p(s_1, z_1) \prod_{t=1}^{H} p(o_t \mid s_t)\, p(a_t \mid h_t)\, p(s_{t+1} \mid s_t, a_t, z_t)\, p(z_{t+1} \mid z_t). \tag{1}$$

Here $p(z_{t+1} \neq z_t) = h$. The regime multiplies plausible next states for the same $(s_t, a_t)$, so a stationary policy can fail after a switch.

**Surprise and hazard for switches.** We signal a switch by failed prediction. Define surprise as the negative log predictive likelihood under the belief-averaged regime model:

$$S_t = -\log \left[ \sum_{z_t} p(o_t \mid h_{t-1}, z_t) \underbrace{\sum_{z_{t-1}} p(z_t \mid z_{t-1})\, q(z_{t-1})}_{\text{predictive regime prior}} \right]. \tag{2}$$

We combine $S_t$ with the hazard in a Bayesian online changepoint detection update over run length (Adams & MacKay, 2007; Itti & Baldi, 2005). A large $S_t$ or a short run-length posterior triggers a reset toward short histories and accelerates regime belief updates.

## 2.2 ACTIVE INFERENCE

Active Inference casts control as approximate inference in Eq. (1). We choose actions that minimize Expected Free Energy (EFE) of an open-loop plan $\pi$:

$$\mathcal{G}(\pi) = \sum_{t=1}^{H} \Big[ \underbrace{\mathbb{E}_q[-\log p(o_t^\star \mid s_t)]}_{\text{risk}} - \lambda \underbrace{I_q(s_t; o_t)}_{\text{epistemic value}} \Big], \tag{3}$$

with expectations under the rollout induced by $\pi$ and current beliefs over $(s_t, z_t)$. We use a light $\lambda$ to induce short probes after suspected switches, then default to goal pursuit.

**Filtering by free energy.** We maintain $q(s_t, z_t)$ with a one-pass variational update that minimizes the per-time variational free energy

$$\mathcal{F}_t(q) = \text{KL}\big(q(s_t, z_t) \,\|\, p(s_t, z_t \mid h_{t-1})\big) - \mathbb{E}_q[\log p(o_t \mid s_t)]. \tag{4}$$

Minimizing (4) yields $q(s_t, z_t) \propto p(o_t \mid s_t)\, p(s_t, z_t \mid h_{t-1})$, i.e., a single-pass filter consistent with Eq. (1). This $q$ feeds EFE and conditions the planner; no rewards are used.

## 2.3 DIFFUSION PLANNING

We use a trajectory diffusion model as a belief-aware control sampler (Sohl-Dickstein et al., 2015; Ho et al., 2020; Dhariwal & Nichol, 2021; Ho & Salimans, 2022).

**Objects.** The plan $x_0 \in \mathbb{R}^{dH}$ stacks $H$ actions. The conditioner $c = (s_t, g, \tilde{z}_t)$ concatenates the current state, a goal embedding for $p(o_t^\star \mid s_t)$, and compact regime-belief features (e.g., $q(z_t)$ and recent run length).

**Training.** We train a conditional denoiser $\epsilon_\theta(x_n, n, c)$ on offline plans with the standard DDPM forward process

$$q(x_n \mid x_{n-1}) = \mathcal{N}(\sqrt{\alpha_n}\, x_{n-1},\, \beta_n I), \tag{5}$$

and the Gaussian reverse model

$$p_\theta(x_{n-1} \mid x_n, c) = \mathcal{N}\big(\mu_\theta(x_n, n, c),\, \tilde{\beta}_n I\big), \quad \mu_\theta = \frac{1}{\sqrt{\alpha_n}} \left( x_n - \frac{\beta_n}{\sqrt{1 - \bar{\alpha}_n}} \epsilon_\theta(x_n, n, c) \right). \tag{6}$$

**Test-time planning.** Initialize $x_N \sim \mathcal{N}(0, I)$ and iterate

$$x_{n-1} = \mu_\theta(x_n, n, c) + \sigma_n\, \xi, \qquad \xi \sim \mathcal{N}(0, I), \tag{7}$$

to sample a multi-modal, temporally coherent plan $x_0$. Execute $a_t$ (the first block), shift the horizon, and replan.

**Belief-aware migration.** When BOCPD shifts $q(z_t)$ after a surprise, $\tilde{z}_t$ changes and so does the conditional score. The induced plan distribution moves from regime (b) toward regime (c) in Fig. 2.

**Classifier-free guidance for goals and probes.** At test time we trade fidelity for directed behavior via

$$\epsilon_{\text{guided}} = \epsilon_\theta(x_n, n, c) + \omega\big(\epsilon_\theta(x_n, n, c) - \epsilon_\theta(x_n, n, \varnothing)\big). \tag{8}$$

We modulate $\omega$ with a light epistemic signal (proportional to the predicted $I_q(s_t; o_t)$ or short run-length posterior) to favor brief information-gathering moves near suspected switches, then return to pure goal pursuit.

## 2.4 Summary

Meta-RL adapts with gradients or memory (Finn et al., 2017; Duan et al., 2017). Context methods infer a task latent online and condition a policy (Rakelly et al., 2019; Zintgraf et al., 2020). Offline RL such as IQL is stable under pooled data (Kostrikov et al., 2022). Our stance: detect switches from surprise with a hazard prior, maintain a regime belief with the free-energy filter (4), and condition a diffusion planner on that belief. A light epistemic drive yields short, informative probes after suspected switches. After a switch, the control distribution should migrate from panel (b) to panel (c) with minimal delay while remaining task-directed.

## 3 Methodology

**AIF-Diffusion** couples online belief inference with diffusion-based trajectory planning under an Expected Free Energy (EFE) objective. The controller operates in closed loop and comprises three interacting parts. First, a surprise-triggered belief updater maintains a compact posterior over latent dynamics. Second, a conditional diffusion model proposes short-horizon action sequences conditioned on the current belief and goal. Third, an EFE-based scorer evaluates each proposal under the agent's predictive model and selects the executed action. Figure and notation follow Section 2. We give a self-contained specification below.

### 3.1 Belief inference and changepoint detection

We assume a latent Markovian state $s_t$, an observation $o_t$, and a discrete regime $z_t \in \{1, \ldots, |\mathcal{Z}|\}$ that indexes the active dynamics (e.g., friction or gravity settings). The agent keeps factorized beliefs

$$q(z_t \mid o_{\leq t}), \qquad q(s_t \mid z_t, o_{\leq t}),$$

which it updates online by combining a one-step prediction with the new likelihood. Let $h_{t-1}$ summarize history up to $t-1$ (controls, states, and sufficient statistics of beliefs). The variational filtering step is

$$q(s_t, z_t) \propto p(o_t \mid s_t)\, p(s_t, z_t \mid h_{t-1}), \tag{9}$$

where the predictive prior $p(s_t, z_t \mid h_{t-1})$ is obtained by pushing the previous belief through the regime transition model and state dynamics (Section 2). This update is the per-step free-energy minimizer and reduces to the standard Bayesian filter when the variational family is exact.

**Surprise and run-length logic.** Before updating, we compute an *instantaneous surprise* that measures how unlikely $o_t$ was under the predictive mixture over regimes:

$$S_t = -\log \left[ \sum_z p(o_t \mid h_{t-1}, z)\, q(z_{t-1}{=}z) \right]. \tag{10}$$

Large $S_t$ indicates that the current model of the world poorly explains the new data.

To respond to abrupt shifts, we also maintain a run-length distribution $p(r_t \mid o_{\leq t})$ (time since the last changepoint) under a constant hazard $h \in (0, 1)$. The prior probability of a change at step $t$ is

$$\Pr(\text{change at } t \mid o_{\leq t}) = \sum_{r_{t-1}} h\, p(r_{t-1} \mid o_{\leq t-1}), \tag{11}$$

which encourages resets at a rate set by $h$. In practice we combine this prior signal with surprise to decide whether to discount stale history.

When either signal indicates a likely shift,

$$S_t > \tau_S \quad \text{or} \quad \Pr(\text{change at } t \mid o_{\leq t}) > \tau_C, \tag{12}$$

we *rapidly reset* by reweighting $p(r_t)$ toward small run-lengths (favoring a new segment) and flattening $q(z_t)$ over regimes. We then apply (9). This mechanism allows within-episode adaptation to sudden changes such as altered friction, mass, or gravity.

## 3.2 DIFFUSION TRAJECTORY PLANNER

Rather than selecting a single action, we plan an open-loop sequence of length $H$, $x_{0:H-1} \equiv (a_t, \ldots, a_{t+H-1})$, and execute only the first control in closed loop. Offline, we train a trajectory-level denoising diffusion model on demonstrations collected across regimes. The model is conditioned on

$$c_t = \begin{bmatrix} s_t, g, \tilde{z}_t \end{bmatrix}, \tag{13}$$

where $g$ encodes preferred outcomes (goals), and $\tilde{z}_t$ is a compact embedding of the regime belief $q(z_t)$, e.g., a temperature-scaled soft one-hot that is passed through a small MLP to produce a fixed-width feature.

At test time we draw $K$ candidate sequences by running reverse-time denoising from Gaussian noise $x_N \sim \mathcal{N}(0, I)$. Let $\epsilon_\theta(x_n, c, n)$ be the learned noise predictor at diffusion step $n$. We apply classifier-free guidance (CFG) to trade off the conditional and unconditional branches:

$$\hat{\epsilon}_\theta(x_n, c, n; \omega_t) = (1 + \omega_t)\, \epsilon_\theta(x_n, c, n) \ - \ \omega_t\, \epsilon_\theta(x_n, \varnothing, n), \tag{14}$$

$$x_{n-1} = \mu_n(x_n, \hat{\epsilon}_\theta) + \sigma_n\, \xi, \quad \xi \sim \mathcal{N}(0, I). \tag{15}$$

The guidance weight $\omega_t$ is set from current epistemic uncertainty in the regime belief:

$$\omega_t = \omega_{\min} + \kappa \, \frac{\mathsf{H}[q(z_t)]}{\log |\mathcal{Z}|} \ \in \ [\omega_{\min}, \omega_{\max}], \tag{16}$$

where $\mathsf{H}$ denotes entropy. Larger $\omega_t$ places more weight on the goal-conditioned branch, which steers proposals toward $c_t$; smaller $\omega_t$ yields more diverse samples. This schedule provides a simple knob for exploration–exploitation that reacts to belief sharpness.

## 3.3 EFE-GUIDED SCORING AND SELECTION

Each candidate open-loop policy $\pi^{(k)} = (a_t^{(k)}, \ldots, a_{t+H-1}^{(k)})$ is scored with the Expected Free Energy (EFE) formulation from Section 2 (Eq. 3), balancing goal-directed behavior against information gain. We adapt the general formulation to our setting:

$$\mathcal{G}(\pi) = \underbrace{\mathbb{E}_{q(o_{t:t+H}|\pi)}\big[ -\log p^\star(o_{t:t+H})\big]}_{\text{risk}} - \lambda \underbrace{\mathbb{E}_{q(o_{t:t+H}|\pi)}\big[I_q(s_{t:t+H}; o_{t:t+H} \mid \pi)\big]}_{\text{epistemic value}}, \tag{17}$$

where expectations are computed by rolling out the current belief-driven dynamics and observation models for $H$ steps under $\pi$, using Monte Carlo samples or analytic moments when available. We select the lowest-EFE proposal and execute its first action:

$$\pi^\star = \arg\min_{\pi^{(k)}} \mathcal{G}(\pi^{(k)}), \qquad a_t \leftarrow \text{first action of } \pi^\star, \tag{18}$$

and replan at $t+1$ using the updated belief. This implements an explicit trade-off between goal attainment and information gain, avoiding ad hoc reward shaping.

## 3.4 OVERALL ALGORITHM

Algorithm 1 summarizes the loop. At each time step we: (i) measure surprise and changepoint probability; (ii) optionally reset; (iii) update the belief via (9); (iv) form the conditioner $c_t$ and set $\omega_t$ via (16); (v) sample $K$ candidate sequences by iterating (14)–(15); (vi) score candidates with (17); and (vii) execute the first action of the best candidate. Because candidates are high-quality, a small $K$ suffices in practice.

**Complexity and practice.** Let $N$ be diffusion steps, $H$ the horizon, and $K$ the number of proposals. Sampling costs $O(KN)$ evaluations of $\epsilon_\theta$ and $O(K)$ draws of Gaussian noise per step. Scoring adds $O(KH)$ belief-consistent rollouts, which are short and reuse cached likelihood terms inside (17). Belief updates scale as $O(|\mathcal{Z}|)$ for the regime posterior plus the cost of the chosen state filter for $q(s_t \mid z_t)$. Typical settings use $K \in [5, 10]$, $H \in [8, 20]$, and $N \in [10, 50]$, with early-stopped denoising and memoized terms to keep latency low.

---

**Algorithm 1** AIF-Diffusion for nonstationary control

---

**Require:** Pretrained diffusion policy $p_\theta(x_{0:H-1} \mid s, g, \tilde{z})$; initial $q(z_1)$; thresholds $(\tau_S, \tau_C)$; hazard $h$.

 1: **for** $t = 1, 2, \ldots$ **do**
 2:     Observe $o_t$.
 3:     Compute surprise $S_t$ via (2) and changepoint probability via (11).
 4:     **if** $S_t > \tau_S$ **or** $\Pr(\text{change}) > \tau_C$ **then**
 5:         Reset run-length prior toward short segments; flatten $q(z_t)$.
 6:     **end if**
 7:     Belief update: apply (9) to obtain $q(s_t, z_t)$ and features $\tilde{z}_t$.
 8:     Form conditioner $c_t = (s_t, g, \tilde{z}_t)$ as in (13).
 9:     Set guidance weight $\omega_t$ from (16).
10:     **Sampling:** draw $K$ candidates $\{x_{0:H-1}^{(k)}\}_{k=1}^K$ by iterating (14)–(15).
11:     **Scoring:** compute $G^{(k)} = \mathcal{G}(\pi^{(k)})$ with (17) under current belief.
12:     Select $k^\star = \arg\min_k G^{(k)}$ and execute $a_t = x_0^{(k^\star)}$.
13: **end for**

---

## 4 EXPERIMENTS

We evaluate AIDIF in a modified, nonstationary MuJoCo HalfCheetah environment (Todorov et al., 2012) to test whether explicit dynamics inference improves control when the system undergoes abrupt changes, and we compare against Implicit Q-Learning (IQL) (Kostrikov et al., 2022) and a Decision Diffuser baseline implemented from a public codebase (Dong et al., 2024; Janner et al., 2022), using identical training data whenever applicable; across runs with the same budget, AIDIF matches or exceeds the diffuser baseline and clearly outperforms IQL under dynamics shift, and an ablation indicates that the inference pathway is the component that yields robust, multimodal plans.

### 4.1 SETUP AND TRAINING

Offline training uses a mixture of two datasets: (i) the D4RL `halfcheetah-medium-v2` dataset (Fu et al., 2020) containing 1M transitions and (ii) an off-dynamics dataset collected with a Soft Actor–Critic (SAC) policy (Haarnoja et al., 2018) trained to the D4RL protocol in a modified simulator. For the off-dynamics dataset, we collect 250k transitions each from four variants: $5\times$ gravity, friction coefficient 0.1, shortened leg segments ($0.8\times$ length), and tighter joint angle limits ($0.9\times$ range), totaling 1M additional transitions. The datasets are balanced through uniform sampling during training, with states and rewards normalized following D4RL conventions. We adopt the ODRL framework for SAC training and logging (Lyu et al., 2024). Our diffusion policy implementation follows the *clean-diffuser* repository (Dong et al., 2024) and is written in PyTorch with PyTorch Lightning for streamlined training (Falcon & team, 2019). All experiments run on a desktop with an Intel i9-12900K CPU and an NVIDIA RTX 3090 GPU; each model is trained for 500,000 gradient steps with batch size 256, and although further training might benefit some baselines, we keep the budget fixed for comparability. All reported results average over 5 random seeds with different initializations.

### 4.2 UNSUPERVISED REGIME INFERENCE

For the unsupervised variant, we augment the model to infer regime structure without ground-truth labels. During offline training, we learn a discrete latent variable model over trajectory segments using a VAE-style architecture. The encoder $q_\phi(z \mid s_{t:t+H}, a_{t:t+H-1})$ maps state-action sequences to a categorical distribution over $|\mathcal{Z}|=8$ latent codes, while the decoder reconstructs dynamics predictions. The latent codes are trained to maximize the evidence lower bound (ELBO), encouraging distinct codes for different dynamics while maintaining temporal consistency through a KL penalty against a sticky prior $p(z_{t+1} \mid z_t)$. At test time, we initialize the regime belief $q(z_t)$ uniformly and update it using the learned encoder on recent trajectory windows. This approach discovers meaningful regime clusters without supervision, though convergence after a switch is slower than the labeled variant due to the need to accumulate sufficient evidence. The unsupervised model uses the same

| Method | Train data | Return@100 | Reward 100–200 | Return@300 |
|---|---|---|---|---|
| AIDIF (label) | Mixed | 390.97 | 236.85 | 1100.56 |
| AIDIF (unsup) | Mixed | 194.16 | 175.41 | 750.00 |
| Decision Diffuser | D4RL | 390.97 | 8.58 | 825.11 |
| Decision Diffuser | Mixed | 352.30 | −29.22 | 456.66 |
| IQL | Mixed | 88.96 | −52.85 | 89.88 |

Table 1: Cumulative return at step 100, reward accrued during steps 100–200, and cumulative return at step 300 (higher is better).

hyperparameters as the labeled version, with an additional encoding window of 10 steps for regime inference.

## 4.3 MAIN RESULTS

We evaluate with a 300-step rollout in which dynamics switch twice without resetting state: default physics for steps 0–99, a sudden gravity increase to $5\times$ at step 100, and a reversion to default at step 200. Unless noted, returns are D4RL-normalized and the 100–200 reward window uses the same normalized units. We choose gravity shifts over friction or morphology changes because they create clear, reproducible performance differences while maintaining stable dynamics within each regime. For AIDIF, we use $K=8$ candidate trajectories, horizon $H=15$, diffusion steps $N=20$, $\lambda=0.1$ for the EFE trade-off, guidance weights $\omega_{\min}=0.5$ and $\omega_{\max}=2.0$, and hazard rate $h=0.02$. The changepoint thresholds are set to $\tau_S=3.0$ (surprise) and $\tau_C=0.1$ (changepoint probability). Figure 3 visualizes the resulting reward traces with change points marked; Table 1 summarizes three complementary metrics that probe pre-shift competence (**Return@100**), in-shift adaptation (**Reward 100–200**), and end-to-end performance (**Return@300**). Before the switch, AIDIF with labels and the Decision Diffuser trained only on D4RL exhibit comparable competence, whereas IQL trails and the unsupervised AIDIF variant sits in between. Immediately after the gravity jump, the curves separate: both AIDIF variants maintain positive reward and begin to recover within the shifted window, while the diffuser baselines hover near zero or turn negative and IQL degrades further. When dynamics revert at step 200, AIDIF continues to climb and consolidates gains, yielding the strongest **Return@300**. We also stress-tested with friction and morphology perturbations; the qualitative ranking is similar but variance increases, so we report gravity as the canonical case due to its stability and reproducibility.

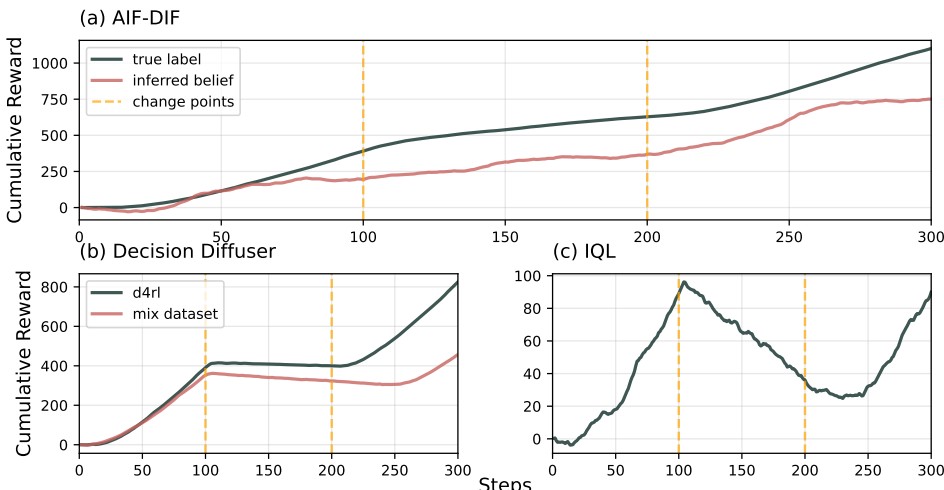

Figure 3: Cumulative reward over time in the nonstationary evaluation; vertical dashed lines indicate the gravity shift at step 100 and the reversion at step 200.

AIDIF with dynamics labels achieves the highest in-shift reward and the best final return, indicating fast belief correction and effective guidance of the diffusion sampler under sudden regime changes. The unsupervised variant, which learns regime structure without ground-truth labels, lags the labeled model yet still surpasses IQL across metrics and remains competitive with the diffuser on **Return@300**, supporting the claim that explicit inference helps even without supervision. Decision Diffuser trained solely on D4RL performs well pre-shift but adapts weakly post-shift; training it on the mixed dataset does not close the gap and can degrade both in-shift reward and final return, consistent with reward-conditioning that entangles task progress and dynamics so the model collapses modes toward the dominant regime. Ablations (not shown in the table) align with this picture: lowering the hazard prior delays resets and hurts **Reward 100–200**; overly aggressive hazard induces spurious resets and costs **Return@100**; disabling EFE guidance during denoising reduces recovery slope; and shrinking the candidate set K narrows the multimodal search and harms robustness. Regarding changepoint detection sensitivity, we find the thresholds $\tau_S$ and $\tau_C$ require careful tuning: values of $\tau_S \in [2.5, 3.5]$ and $\tau_C \in [0.05, 0.15]$ yield stable performance, but outside these ranges the controller either misses switches (high thresholds) or resets spuriously during normal operation (low thresholds). The hazard rate $h$ acts as a prior on switch frequency; we select $h=0.02$ based on expected episode length and typical switch timing, though values in $[0.01, 0.05]$ produce similar results. These parameters could be adapted online in future work based on observed switch patterns. We also observe asymmetric transfer: policies exposed to "hard" regimes (e.g., high gravity) generalize down to easier settings more readily than the reverse, likely due to broader state–action coverage during offline training. Overall, the evidence indicates that coupling online dynamics inference with EFE-guided diffusion planning improves in-shift reward (steps 100–200) and produces reliably higher end-to-end return under nonstationarity, while highlighting headroom to improve unsupervised regime discovery and the immediacy of the first few post-shift actions.

## 5 CONCLUSION AND OUTLOOK

We presented AIDIF, a controller that unifies online belief inference, belief-conditioned diffusion planning, and lightweight Expected Free Energy (EFE) guidance for reward-free adaptation to abrupt, within-episode dynamics shifts. A fast variational filter maintains $q(s_t, z_t)$ under a hazard prior; prediction surprise triggers changepoint resets; a conditional diffusion model trained on regime-stratified data samples $K$ horizon-$H$ trajectories conditioned on $(s_t, g, \tilde{z}_t)$; a tractable EFE proxy both guides denoising and ranks candidates. On nonstationary MuJoCo tasks with mid-episode gravity changes, AIDIF improves in-shift reward (steps 100–200) and preserves final return relative to Decision Diffuser and IQL under matched budgets. The label-supervised variant adapts fastest; the unsupervised variant remains competitive, indicating that belief quality is the critical lever.

**Limitations.** Belief sensitivity: misspecified observation or dynamics models can delay or destabilize $q(z_t)$ updates; discrete $z_t$ and finite regime coverage leave blind spots for coupled or continuous changes; preference design in $p(o^\star! \mid s)$ can bias EFE and induce over-probing; guidance introduces latency and hyperparameters $(K, H, N, \omega, \lambda, h)$ that require tuning; evaluation focuses on one agent and dominant perturbations, so external validity to richer domains and hardware is untested.

**Future work.** Three directions appear most promising for extending this framework. First, *continuous regime inference* could replace our discrete $z_t$ with a continuous latent space, using normalizing flows or VAEs to capture gradual parameter drift alongside abrupt changes—this would address the current limitation where discrete regimes miss intermediate dynamics. Second, *adaptive hyperparameter scheduling* could automatically adjust the changepoint thresholds $(\tau_S, \tau_C)$ and guidance weight $\omega$ based on recent switch patterns, reducing the manual tuning burden identified in our sensitivity analysis. Third, *safety-aware EFE formulations* could incorporate chance constraints or control barrier functions directly into the epistemic value term of Eq. 17, ensuring exploration remains within safe operating envelopes—critical for real-world deployment where our current unconstrained information-seeking could violate safety limits. Each extension builds naturally on our belief-conditioned diffusion architecture while addressing specific limitations observed in the experiments.

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
