# OpenReview forum: "AIF-Guided Diffusion Planning for Nonstationary Control"
_ICLR.cc/2026/Conference — ICLR 2026 Conference Withdrawn Submission_

### Official Review · Reviewer_aPNV · 2025-10-14

**Soundness:** 1
**Presentation:** 1
**Contribution:** 2
**Rating:** 2
**Confidence:** 4

**Summary:**

The paper proposes AIF-Guided Diffusion Planning (AIDIF), which integrates Active Inference (AIF) principles with diffusion-based trajectory planners to handle nonstationary control without reward signals during deployment. The approach uses changepoint detection to trigger belief updates and employs Expected Free Energy (EFE) for plan selection. Experiments on a modified nonstationary MuJoCo HalfCheetah task show faster recovery after dynamics shifts compared to IQL and Decision Diffuser baselines.

**Strengths:**

- Addresses a relevant and important problem: adaptation under nonstationary dynamics without online rewards.
- Presents a conceptually interesting combination of Active Inference’s belief-based reasoning with diffusion-based planners.
- Includes both supervised and unsupervised variants, along with some ablation analysis.
- Offers comprehensive experimental reporting (Return@100, Reward 100–200, Return@300) and explores ablation on hazard rate and guidance weight.

**Weaknesses:**

- Limited novelty: The method is largely a composition of existing frameworks (AIF, DDPM planning, Bayesian changepoint detection) with minimal new theory.
- Weak empirical validation: Results are restricted to a single custom HalfCheetah setup; no tests on other domains or real dynamics shifts, and even D4RL coverage is incomplete.
- Insufficient baselines: Comparisons omit stronger nonstationary or adaptive RL baselines (e.g., PEARL, RMA).
- Reproducibility concerns: Implementation details (EFE computation, hyperparameters, model architecture) are vague or missing.
- Poor clarity: The paper’s presentation and explanations obscure the key technical contributions.
- Incomplete execution: Several aspects suggest the paper is unfinished or rushed — for example, Figure 2 is misaligned/off-page and certain equations lack context or definitions.

**Questions:**

1. How is the Expected Free Energy computed efficiently during denoising — analytically, via sampling, or through approximation?
2. How does the proposed changepoint mechanism differ from standard Bayesian Online Changepoint Detection (Adams & MacKay, 2007)?
3. Could the method generalize beyond MuJoCo gravity shifts, e.g., to morphology or sensor noise changes?
4. Are the unsupervised latent regimes stable across seeds, or do they vary arbitrarily?

Others:
- Could you polish some parts, e.g., lines 092–103, carefully to make them more readable?
- Could you unify the reference format used in equations?
- Could you re-arrange and resize all figures to improve visual quality, to make this paper presentation closer to ICLR standards?

---

### Official Review · Reviewer_2oNk · 2025-11-01

**Soundness:** 2
**Presentation:** 2
**Contribution:** 2
**Rating:** 2
**Confidence:** 4

**Summary:**

The paper considers nonstationary control, and proposes AIDIF, a controller that unifies online belief inference, belief conditioned diffusion planning, and Expected Free Energy (EFE) guidance for reward-free adaptation to abrupt dynamic changes. The method and experiment on MuJoCo HalfCheetah are presented.

**Strengths:**

The strength of the paper comes from the attempt to address nonstationary, especially abrupt dynamics shifts. The proposed method casts controlling as an active inference (AIF) problem and combines VFE, DDPM, EFE in order to tackle rapid and reward-free adaptation.

**Weaknesses:**

### Writing

The paper becomes hard to parse from time to time. For instance, it might worth considering breaking down a long sentence into organized shorter ones (lines 105--107). The material re-organization might help ensure a smooth and clear delivery of intended takeaways. For instance, the appearance of exp detail paragraph (lines 92--103) seems strange in Section 1. The writing occasionally feels like a compilation of bulletin points than a coherent article (line 140, "Objective: ...").

### Motivation and Novelty

While I can understand that the paper aims to address the controlling problem in the presence of abrupt dynamics shifts, the proposed method is more of "combining" different components from previous works than presenting a well-motivated novel solution.

### Experiment

The main experiment is conducted on MuJoCo HalfCheetah with a relatively limited number of "abrupt regime changes." More comprehensive experimental settings would help make the claimed contribution more substantial.

**Questions:**

Can authors clarify the motivation and significance behind the proposed combination of previous-work components?

What are additional practical settings (or datasets) where the proposed approach excels?

---

### Official Review · Reviewer_WC4Z · 2025-11-01

**Soundness:** 2
**Presentation:** 1
**Contribution:** 1
**Rating:** 2
**Confidence:** 3

**Summary:**

This paper proposes a belief-aware diffusion model for control in non-stationary environments. Specifically, it learns a belief model over dynamics using change-point detection (with a hazard prior) to identify when the environment changes, and then conditions the diffusion planner on this belief for control. The objective combines optimal planning with information gain through an expected free energy form. Experiments are conducted on the Half-Cheetah environment, comparing the method to vanilla diffusion planners.

Overall, the problem setting is meaningful and highly relevant for real-world control and RL, where adapting to non-stationary dynamics is crucial. However, in its current form, the paper lacks rigorous modeling of the problem setup and equations, misses important discussions and comparisons with related work, and contains several unclear or incomplete parts in both the method and writing. Given these issues, I find it below the publication bar for now, though the direction itself is promising. I will list my detailed comments below.

**Strengths:**

1. The motivation is clear, and the problem is important for RL as non-stationarity in real-world settings can easily break a fixed policy or model if we don’t explicitly model changes.

2. Although some parts are unclear, the overall modeling idea, such as detecting change points, maintaining a belief over dynamics, and conditioning the diffusion planner on that belief is generally reasonable.

**Weaknesses:**

I list weaknesses and questions together here since they mostly overlap.

1.  Why do we have to stay in the within-episode change setting? The current formulation seems general enough to handle changes at episode start or across episodes, since change-point detection can run over a sequence of episodes as well.


2. The problem is very close to contextual MDPs [1], dynamic latent contextual MDPs [2], Hidden-Parameter MDPs (HiP-MDP) [3], and dynamic HiP-MDPs [4]. It would be better to formalize your setting explicitly in that line of work, so the assumptions and objectives are clearer.


3. Since [2, 4] already maintain beliefs over latent dynamics, you could directly plug their belief models into your diffusion controller and compare under the same diffusion architecture. That would isolate the benefit of your belief update vs. theirs.


4. Meta-learning approaches, especially diffusion-based ones like Meta-Diffuser, should be considered as baselines. Even if the setup is slightly different, adapting on a few contexts and then planning with diffusion is close enough to warrant a comparison.


5. At the moment, it’s a bit unclear what the core novelty is over [2]. They also do latent belief learning, change-point-style adaptation, and plan under uncertainty. Even the “risk + epistemic/expected information gain” view is standard in this area. It would help to make the extra benefit of your objective explicit.


6. The experiments could be stronger — adding more non-stationary benchmarks (e.g. those used in Meta-Diffuser and in [2]) would make the empirical story more convincing.


7. The writing can be improved: many acronyms (POMDP, BOCPD, etc.) are used before being defined; some of Sec. 2 is redundant; and the flow is a bit hard to follow.


8. Please list the hyperparameters for the belief model and the diffusion planner  (I couldn’t find them for now).


[1] Hallak, Assaf, Dotan Di Castro, and Shie Mannor. "Contextual markov decision processes." arXiv preprint arXiv:1502.02259 (2015).

[2] Liang, Anthony, et al. "DynaMITE-RL: A dynamic model for improved temporal meta-reinforcement learning." Advances in Neural Information Processing Systems 37 (2024): 141390-141416.

[3] Doshi-Velez, Finale, and George Konidaris. "Hidden parameter markov decision processes: A semiparametric regression approach for discovering latent task parametrizations." IJCAI: proceedings of the conference. Vol. 2016. 2016.

[4] Xie, Annie, James Harrison, and Chelsea Finn. "Deep reinforcement learning amidst lifelong non-stationarity." arXiv preprint arXiv:2006.10701 (2020).

**Questions:**

Please find the questions in the above section.

---

### Official Review · Reviewer_6DKM · 2025-11-01

**Soundness:** 3
**Presentation:** 3
**Contribution:** 3
**Rating:** 6
**Confidence:** 3

**Summary:**

This paper introduces AIF-guided diffusion planning, a novel framework for continuous control under abrupt, within-episode dynamics shifts. The core contribution is a system that couples online belief inference over latent dynamics modes ($z_t$) with a belief-conditioned diffusion planner. At test time, the controller uses prediction surprise to detect changepoints and leverages a lightweight Expected Free Energy (EFE) objective to both guide trajectory generation and select the final action. This design enables rapid adaptation to unforeseen changes without requiring online reward signals.

**Strengths:**

- The paper creatively synthesizes principles from Active Inference with modern diffusion-based planning. Using Expected Free Energy as a principled, reward-free guidance and selection mechanism for a powerful generative model is a novel contribution.
- The work is of good significance, tackling the crucial real-world challenge of nonstationarity in control.

**Weaknesses:**

- The empirical validation is confined to a single environment (HalfCheetah), as explicitly stated in lines 344 and 472. This narrow scope may limit the generalizability of the findings, as the method's effectiveness on agents with different morphologies and more complex stability requirements remains unevaluated.
- Second, the method exhibits high sensitivity to key hyperparameters. The authors acknowledge that the changepoint detection thresholds ($\tau_S, \tau_C$) require "careful tuning" (line 444), suggesting that deploying the system to new scenarios may demand considerable manual effort, which potentially undermines its practical utility.

**Questions:**

- What is the typical wall-clock inference time per step for AIDIF compared to the baselines? This information is vital for understanding the method's feasibility for real-time control applications.
- The unsupervised variant shows a notable performance gap compared to the supervised one. Does this suggest a fundamental limitation in discovering meaningful dynamics from offline data alone in real-world scenarios, and how might this gap be closed in future work?

---

### Note · Authors · 2025-12-01

I have read and agree with the venue's withdrawal policy on behalf of myself and my co-authors.